# L3A: Label-Augmented Analytic Adaptation for Multi-Label Class Incremental Learning

Xiang Zhang [1]   Run He [1]   Jiao Chen [1]   Di Fang [2]
Ming Li [3]   Ziqian Zeng [1]   Cen Chen [2]   Huiping Zhuang [1]

## Abstract

Class-incremental learning (CIL) enables models to learn new classes continually without forgetting previously acquired knowledge. Multi-label CIL (MLCIL) extends CIL to a real-world scenario where each sample may belong to multiple classes, introducing several challenges: label absence, which leads to incomplete historical information due to missing labels, and class imbalance, which results in the model bias toward majority classes. To address these challenges, we propose Label-Augmented Analytic Adaptation (L3A), an exemplar-free approach without storing past samples. L3A integrates two key modules. The pseudo-label (PL) module implements label augmentation by generating pseudo-labels for current phase samples, addressing the label absence problem. The weighted analytic classifier (WAC) derives a closed-form solution for neural networks. It introduces sample-specific weights to adaptively balance the class contribution and mitigate class imbalance. Experiments on MS-COCO and PASCAL VOC datasets demonstrate that L3A outperforms existing methods in MLCIL tasks. Our code is available at https://github.com/scut-zx/L3A.

## 1. Introduction

Class-incremental learning (CIL) enables models to acquire knowledge of new classes in a phase-by-phase manner, while maintaining knowledge of previously learned classes without retraining the model from scratch. Research

[1]Shien-Ming Wu School of Intelligent Engineering, South China University of Technology, Guangzhou, China [2]School of Future Technology, South China University of Technology, Guangzhou, China [3]Guangdong Laboratory of Artificial Intelligence and Digital Economy (SZ). Correspondence to: Huiping Zhuang <hpzhuang@scut.edu.cn>.

*Proceedings of the 42^{nd} International Conference on Machine Learning*, Vancouver, Canada. PMLR 267, 2025. Copyright 2025 by the author(s).

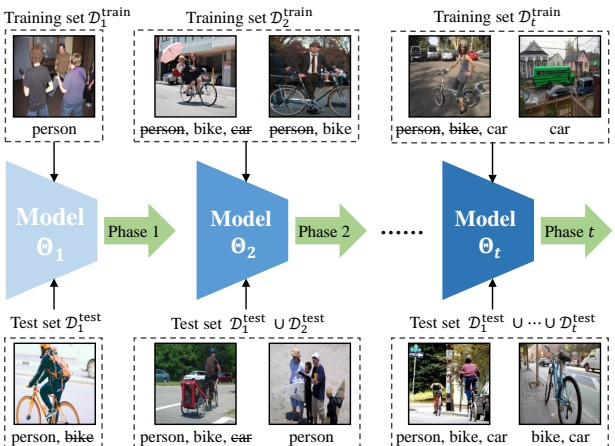

*Figure 1.* The process of MLCIL. The model learns different classes such as 'person', 'bicycle', and 'car' during phases 1, 2, and $t$. '~~person~~' represents samples that belong to this class but lack the label. Different colors of Model $\Theta_1$, $\Theta_2$, and $\Theta_t$ represent the changes in model parameters caused by continual learning.

on CIL faces the severe catastrophic forgetting problem (McCloskey & Cohen, 1989; Ratcliff, 1990), manifested as performance degradation of previously learned classes. Existing CIL methods (Zhou et al., 2024b) have achieved considerable success in single-label CIL tasks, significantly reducing the computational resources for model retraining.

However, in complex real-world environments, a CIL agent needs to learn complex training data with more than one object in one image, termed multi-label CIL (MLCIL). Figure 1 illustrates the detail paradigm, where each training sample is labeled only with the classes relevant to the current learning phase, while the labels of other classes are masked. The goal of MLCIL is to enable the model to eventually recognize all classes encountered throughout the learning process, even if some labels are masked during training phases.

MLCIL faces three major challenges: (1) *label absence*: each sample is label-incomplete, containing only the labels from the current phase, with the labels from previous and future phases absent. For example, the left sample in the training set $\mathcal{D}_t^{\text{train}}$ in Figure 1 includes a person riding a bicy-

cle in front of cars. While the sample contains three classes, it only annotated the 'car' as a training label, with 'person' and 'bike' being part of the background. This causes more severe catastrophic forgetting, as the current samples become negative samples of previously learned labels (Dong et al., 2023). (2) *class imbalance*: the strong co-occurrence of classes in multi-label images leads certain classes to appear together frequently. For example, images with bicycles often include people. This creates an imbalanced class distribution, causing the model to favor high-frequency classes and neglect rare ones, leading to degraded performance. (3) *privacy protection*: privacy constraints often prevent access to historical sample data in real-world scenarios, rendering many replay-based methods inapplicable.

Existing CIL methods fail to address all three challenges of MLCIL. Although many single-label CIL approaches mitigate catastrophic forgetting, they are limited by label absence and are unable to handle the MLCIL problem effectively. To address label absence, KRT (Dong et al., 2023) proposes a knowledge restore and transfer framework, and CSC (Du et al., 2025) introduces a class-incremental graph convolutional network to establish label relationships. Both methods rely on replaying samples to achieve optimal performance. MULTI-LANE (De Min et al., 2024) utilizes a pre-trained large model and prompt-tuning techniques to achieve replay-free MLCIL. However, these MLCIL methods fail to address the class imbalance issue and result in suboptimal performance.

Considering privacy protection scenarios, the exemplar-free CIL (EFCIL) methods without storing historical examples are becoming increasingly important. However, many EF-CIL methods suffer from task-recency bias due to the backpropagation algorithm, where the model focuses on optimizing the most recent tasks and leads to poor performance on previous tasks. Analytic Continual Learning (ACL) (Zhuang et al., 2022) is a new branch of EFCIL that achieves superior performance compared to replay-based methods. ACL enables incremental learning by iteratively solving a closed-form solution, which obtains an equivalent solution to joint training without storing historical data. This characteristic makes ACL well-suited for protecting privacy. However, existing ACL methods are primarily effective in single-label CIL, they encounter limitations when addressing the label absence and class imbalance problem in MLCIL.

To overcome these limitations, we propose the Label-Augmented Analytic Adaptation for Multi-Label Class-Incremental Learning (L3A). Our approach comprises two main components: (1) the pseudo-label (PL) module and (2) the weighted analytic classifier (WAC). The PL module generates label information for previously learned classes by inputting new samples into the old classifier, enabling analytic learning to derive optimal closed-form solutions.

The WAC updates the classifier by recursively solving a ridge regression problem while introducing sample-specific weights to balance the loss contributions of different classes. Our contributions are summarized as follows:

- We propose L3A, an exemplar-free approach that provides a closed-form solution to address catastrophic forgetting in MLCIL.

- We introduce the PL module to implement label-augmented by generating labels for previously learned classes, addressing the label absence problem.

- We introduce the WAC that iteratively updates the classifier by analytic learning and adaptively assigns sample-specific weights, solving the class imbalance problem.

- Experiments on MS-COCO and PASCAL VOC datasets demonstrate that our approach achieves state-of-the-art (SOTA) performance in MLCIL tasks.

## 2. Related Works

### 2.1. Single-Label CIL

**Replay-based** CIL methods such as iCaRL (Rebuffi et al., 2017), ER (Rolnick et al., 2019), BiC (Wu et al., 2019), DER++ (Buzzega et al., 2020), TPCIL (Tao et al., 2020), PODNet (Douillard et al., 2020) mitigate catastrophic forgetting by revisiting old task data during training new task. Although the replay-based methods are straightforward and effective approaches, they face significant challenges under stricter privacy protection constraints.

**Regularization-based** CIL methods constrain certain model parameters to minimize changes to those critical for old task classification. EWC (Kirkpatrick et al., 2017), oEWC (Schwarz et al., 2018) and RWalk (Chaudhry et al., 2018) introduce regularization into the loss function and evaluate the importance of model parameter using the Fisher Information Matrix. LwF (Li & Hoiem, 2017) introduces knowledge distillation (Hinton et al., 2015) into the loss function to preserve old knowledge. However, those models can not achieve better performance compared to replay-based methods commonly.

**Prompt-based** CIL methods utilize pre-trained large models as backbone networks to enhance feature representation ability. L2P (Wang et al., 2022) and CODA-Prompt (Smith et al., 2023) introduce prompt tuning into continual learning, EASE (Zhou et al., 2024a) makes a task-specific subspace for each incremental task.

### 2.2. Multi-Label CIL

To address the label absence, class imbalance, and privacy protection problem in MLCIL, several approaches are pro-

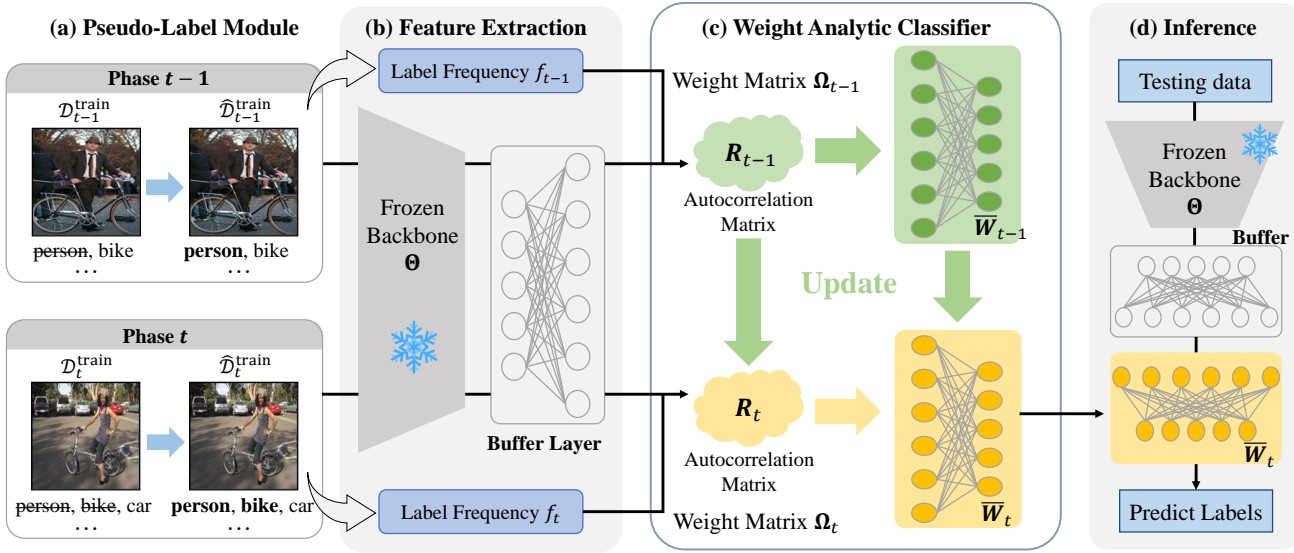

*Figure 2.* The overview of our L3A method. (a) The multi-label data stream arrives in phase, with each sample containing incomplete label information (e.g., '~~person~~'). The pseudo-label module generates the augmented training set $\hat{\mathcal{D}}_t^{\text{train}}$ (e.g., '**person**'). (b) A frozen backbone with a buffer layer extracts sample features and projects them into a higher-dimensional space. (c) The weight analytic classifier iteratively updates the weight matrix $\mathbf{\Omega}_t$, autocorrelation matrix $\boldsymbol{R}_t$, and analytic classifier $\bar{\boldsymbol{W}}_t$ across different phases. (d) The frozen backbone with a buffer layer and the current phase classifier are used for inference.

posed. PRS (Kim et al., 2020) adopts a replay-based strategy to tackle intra- and inter-task class imbalance. OCDM (Liang & Li, 2022) introduces a greedy algorithm to manage class distribution in memory effectively. KRT (Dong et al., 2023) incorporates incremental cross-attention modules to restore and transfer knowledge between continual learning stages. MULTI-LANE (De Min et al., 2024) integrates prompt-based techniques into MLCIL, enabling an exemplar-free MLCIL paradigm. CSC (Du et al., 2025) establishes cross-task label relationships and calibrates the model's over-confident output.

## 2.3. Analytic Continual Learning (ACL)

ACL is an emerging paradigm in continual learning, inspired by pseudo-inverse learning (Guo & Lyu, 2004). It implements an exemplar-free approach that uses recursive least squares to find a closed-form solution for network parameters. ACIL (Zhuang et al., 2022) demonstrates that using recursive least squares is equivalent to the joint learning approach firstly, enabling the application of analytic learning to the field of CIL. DS-AL (Zhuang et al., 2024b) introduces another linear classifier to improve the under-fitting limitation. REAL (He et al., 2024) introduces a representation enhancing distillation process to strengthen the representation of the feature extractor. GACL (Zhuang et al., 2024a) extends the ACL to generalized CIL, dealing with the mixed data categories and unknown sample size distribution. AIR (Fang et al., 2024) addresses the class imbalance in single-label CIL by introducing a re-weighting module.

## 3. Method

### 3.1. Problem Definition

The goal of MLCIL is for a model to incrementally learn the classification of multi-labels within input samples, while the labels introduced at each training phase are incomplete.

Let the phases as $\{\mathcal{D}_1, \mathcal{D}_2, \cdots, \mathcal{D}_t, \cdots\}$, $\mathcal{D}_t$ is divided into a training set $\mathcal{D}_t^{\text{train}}$ and a test set $\mathcal{D}_t^{\text{test}}$. $\mathcal{D}_t^{\text{train}} = \{(\boldsymbol{\mathcal{X}}_{t,1}, \boldsymbol{y}_{t,1}), \cdots, (\boldsymbol{\mathcal{X}}_{t,i}, \boldsymbol{y}_{t,i}), \cdots, (\boldsymbol{\mathcal{X}}_{t,N_t}, \boldsymbol{y}_{t,N_t})\}$ of size $N_t$, where $\boldsymbol{\mathcal{X}}_{t,i}$ represents a input samples tensor, $\boldsymbol{y}_{t,i}$ represents the corresponding multi-hot labels vector. Each label vector $\boldsymbol{y}_{t,i} \in C^t$, and $C^t$ is the partial label space set specific for phase $t$, with the constraint that $\forall m, n(m \neq n), C^m \cap C^n = \varnothing$. $|C^t|$ is the number of unique classes in phase $t$. The model is trained on $\mathcal{D}_t^{\text{train}}$, and evaluated the performance on $\mathcal{D}_{1:t}^{\text{test}}$, which defined as joint test sets $\mathcal{D}_1^{\text{test}} \cup \cdots \cup \mathcal{D}_t^{\text{test}}$. The cumulative label space for testing expands incrementally and is defined as $C^{1:t} = C^1 \cup \cdots \cup C^t$.

### 3.2. Proposed Framework

Our proposed Label-Augmented Analytic Adaptation (L3A) framework comprises three components: (a) a pseudo-label module that completes missing label information by generating pseudo-labels for historical classes, (b) a feature extraction module that extracts and transforms input samples into high-dimensional feature representations, and (c) a weighted analytic classifier employs a closed-form solution with sample-specific weighting to mitigate class imbalance,

with iterative updates. The detailed architecture of the L3A framework is depicted in Figure 2.

Unlike other methods that rely on backpropagation for model updates, the L3A framework can be formulated as an optimization problem. During the $t$-th phase of MLCIL, the objective is to solve for the linear classifier $W_t$ as follows:

$$\underset{W_t}{\mathrm{argmin}} \, \|\Omega_{1:t}^{1/2}(\phi(\mathcal{X}_{1:t}, \Theta)W_t - f_{\mathrm{PL}}(Y_{1:t}))\|_{\mathrm{F}}^2 + \gamma\|W_t\|_{\mathrm{F}}^2, \tag{1}$$

where $\Omega_{1:t}$ denotes the sample-specific weight matrix. $\phi(\cdot, \Theta)$ represents the feature extractor function with $\Theta$ as the backbone. $\mathcal{X}_{1:t}$ and $Y_{1:t}$ are the input sample matrix and the true labels set from phase 1 to $t$. $f_{\mathrm{PL}}(\cdot)$ refers to the pseudo-label algorithm. $\|\cdot\|_{\mathrm{F}}$ indicates the Frobenius norm and $\gamma$ is the regularization term.

### 3.3. Pseudo-Label Module

The problem of label absence is similar to challenges encountered in semi-supervised learning, where the labels of samples from both past and future phases are missing. Motivated by this approach, we propose the pseudo-label (PL) method to address the label absence problem.

Specifically, in the phase $t$, previous classifier $\bar{W}_{t-1}$ already retains knowledge of historical classes. We utilize this classifier with the current training set $\mathcal{D}_t^{\mathrm{train}} = \{(\mathcal{X}_t, Y_t)\}$ to generate the pseudo-label set $\tilde{Y}_t$ for previous classes. A confidence threshold $\eta \in (0, 1)$ activates pseudo-labels as follows:

$$\tilde{y}_{t,i}^{(k)} = \begin{cases} 1 & \text{if } p_{t,i}^{(k)} \geq \eta \\ 0 & \text{otherwise}, \end{cases} \tag{2}$$

where $k$ indexes classes IDs, and $p_{t,i}^{(k)}$ denotes the predication score for the $k$-th class. The pseudo-label algorithm $f_{\mathrm{PL}}(\cdot)$ results the augmented label set is defined as:

$$\hat{Y}_t = \tilde{Y}_t \cup Y_t. \tag{3}$$

The augmented training set $\hat{\mathcal{D}}_t^{\mathrm{train}} = \{(\mathcal{X}_t, \hat{Y}_t)\}$, which ensures the model incorporates historical class information while retaining the label set for the current phase. Section 4.4 analyses why the pseudo-label module works in our method.

### 3.4. Feature Extraction

To calculate the solution of the classifier, we first obtain the feature representation of the input sample. The function $\phi(\cdot, \Theta)$ extracts the features from the input samples and applies a buffer layer $f_{\mathrm{buffer}}(\cdot)$, mapping features to a higher-dimensional space, which enhances the representational capacity. This process results in the high-dimensional feature expansion matrix $X$, given by

$$X_t \leftarrow \phi(\mathcal{X}_t, \Theta). \tag{4}$$

$\Theta$ indicates the frozen backbone weight and the $f_{\mathrm{buffer}}(\cdot)$ improves feature representation and captures more complex information by employing various mapping strategies (Zhuang et al., 2022; 2023). We implement the $f_{\mathrm{buffer}}(\cdot)$ as a random linear projection layer followed by a non-linear activation function, i.e. $f_{\mathrm{buffer}}(X) = \mathrm{ReLU}(XW_{\mathrm{rand}})$. The weights of projection layer $W_{\mathrm{rand}}$ are randomly initialized.

### 3.5. Weighted Analytic Classifier

In this section, we calculate the solution of the linear classifier in Equation (1). Let $X_t$ be the feature expansion matrix obtained from Section 3.4, and $\hat{Y}_t$ be the corresponding augmented label set obtained from Section 3.3. So $X_{1:t}$ and $\hat{Y}_{1:t}$ indicate the feature matrix and augmented label matrix form 1 to $t$ phase.

$$X_{1:t} = \begin{bmatrix} X_{1:t-1} \\ X_t \end{bmatrix}, \quad \hat{Y}_{1:t} = \begin{bmatrix} \hat{Y}_{1:t-1} & 0 \\ \tilde{Y}_t & Y_t \end{bmatrix}.$$

Unlike traditional CIL methods that rely on backpropagation, ACL methods use ridge regression to derive the analytical expression for the classifier. Therefore, we refer to the corresponding linear classifier as the Analytic Classifier (AC). In the general case, the loss function for the analytic classifier $W_t$ is:

$$\underset{W_t}{\mathrm{argmin}} \, \|X_{1:t}W_t - \hat{Y}_{1:t}\|_{\mathrm{F}}^2 + \gamma\|W_t\|_{\mathrm{F}}^2. \tag{5}$$

In Equation (5), each sample contributes the overall loss equally. In class imbalance scenarios, dominant classes with larger quantities introduce bias into the classifier, which tends to overfit the majority classes while underperforming on the minority classes.

To address the class imbalance problem, we propose the weighted analytic classifier (WAC), which assigns a sample-specific weight $\omega_{t,i}$ based on the frequency of each class in the training data. For a class $k$ with frequency $f^{(k)}$, the class-specific weight is computed as $v_t^{(k)} = 1/\sqrt{f^{(k)}}$, ensuring that minority classes contribute more significantly to the loss. For a sample $(\mathcal{X}_{t,i}, \hat{y}_{t,i})$, the sample-specific weight $\omega_{t,i}$ is the average of the weights $v_t^{(k)}$ for all active classes in $\hat{y}_{t,i}$:

$$\omega_{t,i} = \frac{\sum_{k=1}^{K} \hat{y}_{t,i}^{(k)} \cdot v_t^{(k)}}{\sum_{k=1}^{K} \hat{y}_{t,i}^{(k)}}, \tag{6}$$

where $\omega_{t,i}$ denotes the weight corresponding to the $i$-th sample from the $t$-th phase. This weighting mechanism ensures balanced contributions across classes, mitigating bias toward dominant classes and improving the classifier's robustness. To express the weighting mechanism more concisely, we reformulate it as a matrix representation. Specifically, $\Omega_{1:t}$ is defined as a diagonal matrix that represents the sample-specific weight matrix for all samples across the phases from

1 to $t$:

$$\boldsymbol{\Omega}_{1:t} = \mathrm{diag}(\boldsymbol{\Omega}_1, \boldsymbol{\Omega}_2, \cdots, \boldsymbol{\Omega}_t)$$
$$= \mathrm{diag}(\omega_{1,1}, \cdots, \omega_{2,1}, \cdots, \omega_{t,N_t}). \quad (7)$$

When the weighting mechanism is incorporated into the loss function Equation (5), the original formulation can be rewritten as follows:

$$\underset{\boldsymbol{W}_t}{\arg\min} \|\boldsymbol{\Omega}_{1:t}^{1/2}(\boldsymbol{X}_{1:t}\boldsymbol{W}_t - \hat{\boldsymbol{Y}}_{1:t})\|_{\mathrm{F}}^2 + \gamma\|\boldsymbol{W}_t\|_{\mathrm{F}}^2, \quad (8)$$

which is equal to the optimization problem Equation (1). $\boldsymbol{\Omega}_{1:t}^{1/2}$ denotes the element-wise square root of this diagonal matrix, where each diagonal element is defined as the square root of the corresponding weight.

To minimize the loss function of Equation (8), we compute its derivate with respect to $\boldsymbol{W}_t$:

$$\frac{\partial}{\partial \boldsymbol{W}_t} \left( \|\boldsymbol{\Omega}_{1:t}^{1/2}(\boldsymbol{X}_{1:t}\boldsymbol{W}_t - \hat{\boldsymbol{Y}}_{1:t})\|_{\mathrm{F}}^2 + \gamma\|\boldsymbol{W}_t\|_{\mathrm{F}}^2 \right)$$
$$= 2\boldsymbol{X}_{1:t}^{\top}\boldsymbol{\Omega}_{1:t}\boldsymbol{X}_{1:t}\boldsymbol{W}_t - 2\boldsymbol{X}_{1:t}^{\top}\boldsymbol{\Omega}_{1:t}\hat{\boldsymbol{Y}}_{1:t} + 2\gamma\boldsymbol{W}_t. \quad (9)$$

Let the derivate equal to zero, we can get the closed-form solution of $\bar{\boldsymbol{W}}_t$:

$$\bar{\boldsymbol{W}}_t = (\boldsymbol{X}_{1:t}^{\top}\boldsymbol{\Omega}_{1:t}\boldsymbol{X}_{1:t} + \gamma\boldsymbol{I})^{-1}\boldsymbol{X}_{1:t}^{\top}\boldsymbol{\Omega}_{1:t}\hat{\boldsymbol{Y}}_{1:t}. \quad (10)$$

The goal of L3A is to utilize $\bar{\boldsymbol{W}}_{t-1}$ to update $\bar{\boldsymbol{W}}_t$ recursively without other historical samples or data. We define the autocorrelation matrix in the $t$ phase:

$$\boldsymbol{R}_t = (\boldsymbol{X}_{1:t}^{\top}\boldsymbol{\Omega}_{1:t}\boldsymbol{X}_{1:t} + \gamma\boldsymbol{I})^{-1}. \quad (11)$$

The iterative analytic solution process in WAC is shown in Theorem 3.1.

**Theorem 3.1.** *The optimal solution of $\bar{\boldsymbol{W}}_t$ can be calculated recursively by*

$$\bar{\boldsymbol{W}}_t = \left[ \bar{\boldsymbol{W}}_{t-1} - \boldsymbol{R}_t\boldsymbol{X}_t^{\top}\boldsymbol{\Omega}_t\boldsymbol{X}_t\bar{\boldsymbol{W}}_{t-1} \quad \boldsymbol{R}_t\boldsymbol{X}_t^{\top}\boldsymbol{\Omega}_t\hat{\boldsymbol{Y}}_t \right], \quad (12)$$

*where the autocorrelation matrix can be calculated recursively by*

$$\boldsymbol{R}_t = \boldsymbol{R}_{t-1} - \boldsymbol{R}_{t-1}\boldsymbol{X}_t^{\top}\left(\boldsymbol{\Omega}_t^{-1} + \boldsymbol{X}_t\boldsymbol{R}_{t-1}\boldsymbol{X}_t^{\top}\right)^{-1}\boldsymbol{X}_t\boldsymbol{R}_{t-1}. \quad (13)$$

*Proof.* The proof is provided in Section A. $\quad\square$

Theorem 3.1 represents the update of classifier $\bar{\boldsymbol{W}}_t$ at phase $t$ relies solely on the previous classifier $\bar{\boldsymbol{W}}_{t-1}$ and the current phase data. The solution in Equation (12) is equivalent to the joint-learning formulation presented in Equation (10). The autocorrelation matrix $\boldsymbol{R}_t$ assists in this computation. As shown in Equation (11), it stores knowledge with the

inverse of feature matrix product from past phase samples. However, it is impossible to reverse-engineer the information of individual samples from $\boldsymbol{R}_t$, so the L3A method retains $\boldsymbol{R}_t$ instead of storing any historical samples, achieving exemplar-free learning.

Algorithm 1 shows the pseudo-code of L3A, which utilizes the PL module to generate overall labels, extracts the sample features, and recursively updates the classifier by WAC.

---

**Algorithm 1** Training process of L3A

**Input:** Training set $\mathcal{D}_1^{\mathrm{train}}, \cdots, \mathcal{D}_T^{\mathrm{train}}$ with $\mathcal{D}_t^{\mathrm{train}} \sim (\boldsymbol{\mathcal{X}}_t, \boldsymbol{Y}_t)$, the frozen backbone $\boldsymbol{\Theta}$.
**Initialization:** $\boldsymbol{R}_0 \leftarrow \gamma\boldsymbol{I}$, $\bar{\boldsymbol{W}}_0 \leftarrow 0$
**for** phase $t = 1$ to $T$ **do**
$\quad \triangleright$ *Pseudo-label module.* $\qquad\qquad\qquad\quad \triangleleft$
$\quad \hat{\mathcal{D}}_t^{\mathrm{train}} \sim (\boldsymbol{\mathcal{X}}_t, \hat{\boldsymbol{Y}}_t) \leftarrow f_{\mathrm{PL}}(\mathcal{D}_t^{\mathrm{train}}, \eta, \boldsymbol{\Theta}, \bar{\boldsymbol{W}}_{t-1})$
$\quad \triangleright$ *Count the labels frequency.* $\qquad\qquad\quad \triangleleft$
$\quad \boldsymbol{f}_t \leftarrow \mathrm{Count}(\boldsymbol{Y}_t, \boldsymbol{f}_{t-1})$
$\quad \triangleright$ *Calculate the sample-specific weight.* $\quad \triangleleft$
$\quad$ **for all** $(\boldsymbol{\mathcal{X}}_{t,i}, \hat{\boldsymbol{y}}_{t,i}) \in \hat{\mathcal{D}}_t^{\mathrm{train}}$ **do**
$\quad\quad \omega_{t,i} \leftarrow \mathrm{Weight}(\boldsymbol{f}_t, \hat{\boldsymbol{y}}_{t,i})$
$\quad \boldsymbol{\Omega}_t \leftarrow \mathrm{diag}(\omega_{t,1}, \omega_{t,2}, \cdots, \omega_{t,N_t})$
$\quad \triangleright$ *Feature extraction.* $\qquad\qquad\qquad\quad \triangleleft$
$\quad \boldsymbol{X}_t \leftarrow \phi(\boldsymbol{\mathcal{X}}_t, \boldsymbol{\Theta})$
$\quad$ Update $\boldsymbol{R}_t$ with Equation (13)
$\quad$ Update $\bar{\boldsymbol{W}}_t$ with Equation (12)
**Return:** $\bar{\boldsymbol{W}}_t$

---

## 4. Experiments

### 4.1. Setting

#### 4.1.1. DATASETS

We follow previous works (Dong et al., 2023; De Min et al., 2024) in MLCIL and evaluate our method on MS-COCO 2014 (Lin et al., 2014) and PASCAL VOC 2007 (Everingham et al., 2010) datasets. MS-COCO dataset contains 120k images with 80 category labels, PASCAL VOC dataset contains 10k images with 20 category labels. Those datasets are suitable for MLCIL, but they both exhibit issues with class imbalance. Following previous research (Dong et al., 2023), we evaluate our method using the following protocols on both MS-COCO and PASCAL VOC datasets: (1) *MS-COCO B0-C10*: the model is trained across all 80 classes, divided into 8 continual learning phases, each learning 10 classes. (2) *MS-COCO B40-C10*: the model is initially trained on 40 classes, followed by 4 incremental learning phases on the remaining 40 classes. (3) *VOC B0-C4*: the model is trained on all 20 classes across 5 continual learning phases, with 4 classes learned in each phase. (4) *VOC B10-C2*: the model has a base training on 10 classes, with the subsequent 10 classes trained continually over 5 phases. The order of incremental training is the lexicographical order of

*Table 1.* Comparison of the results among L3A and other methods on MS-COCO dataset. Memory represents the number of replay samples, where 0 indicates that the method is examplar-free. Data **in bold** represents the **best** results and data underline represents the second-best results.

| Method | Type | Memory | MS-COCO B0-C10 | | | | MS-COCO B40-C10 | | | |
| | | | Avg. | Last | | | Avg. | Last | | |
| | | | mAP | CF1 | OF1 | mAP | mAP | CF1 | OF1 | mAP |
| Upper-bound | Baseline | 0 | - | 76.4 | 79.4 | 81.8 | - | 76.4 | 79.4 | 81.8 |
| FT (Ridnik et al., 2021) | Baseline | | 38.3 | 6.1 | 13.4 | 16.9 | 35.1 | 6.0 | 13.6 | 17.0 |
| TPCIL (Tao et al., 2020) | CIL | | 63.8 | 20.1 | 21.6 | 50.8 | 63.1 | 25.3 | 25.1 | 53.1 |
| PODNet (Douillard et al., 2020) | CIL | | 65.7 | 13.6 | 17.3 | 53.4 | 65.4 | 24.2 | 23.4 | 57.8 |
| DER++ (Buzzega et al., 2020) | CIL | 5/class | 68.1 | 33.3 | 36.7 | 54.6 | 69.6 | 41.9 | 43.7 | 59.0 |
| KRT-R (Dong et al., 2023) | MLCIL | | 75.8 | 60.0 | 61.0 | 68.3 | 78.0 | 66.0 | 65.9 | 74.3 |
| CSC-R (Du et al., 2025) | MLCIL | | 79.2 | 67.3 | 68.1 | 73.7 | 78.4 | 68.1 | 69.0 | 75.6 |
| iCaRL (Rebuffi et al., 2017) | CIL | | 59.7 | 19.3 | 22.8 | 43.8 | 65.6 | 22.1 | 25.5 | 55.7 |
| BiC (Wu et al., 2019) | CIL | | 65.0 | 31.0 | 38.1 | 51.1 | 65.5 | 38.1 | 40.7 | 55.9 |
| ER (Rolnick et al., 2019) | CIL | | 60.3 | 40.6 | 43.6 | 47.2 | 68.9 | 58.6 | 61.1 | 61.6 |
| TPCIL (Tao et al., 2020) | CIL | | 69.4 | 51.7 | 52.8 | 60.6 | 72.4 | 60.4 | 62.6 | 66.5 |
| PODNet (Douillard et al., 2020) | CIL | 20/class | 70.0 | 45.2 | 48.7 | 58.8 | 71.0 | 46.6 | 42.1 | 64.2 |
| DER++ (Buzzega et al., 2020) | CIL | | 72.7 | 45.2 | 48.7 | 63.1 | 73.6 | 51.5 | 53.5 | 66.3 |
| KRT-R (Dong et al., 2023) | MLCIL | | 76.5 | 63.9 | 64.7 | 70.2 | 78.3 | 67.9 | 68.9 | 75.2 |
| CSC-R (Du et al., 2025) | MLCIL | | 79.6 | 67.8 | 68.6 | 74.8 | 78.7 | 68.2 | 69.4 | 76.7 |
| PRS (Kim et al., 2020) | MLCIL | | 48.8 | 8.5 | 14.7 | 27.9 | 50.8 | 9.3 | 15.1 | 33.2 |
| OCDM (Liang & Li, 2022) | MLCIL | | 49.5 | 8.6 | 14.9 | 28.5 | 51.3 | 9.5 | 15.5 | 34.0 |
| KRT-R (Dong et al., 2023) | MLCIL | 1000 | 75.7 | 61.6 | 63.6 | 69.3 | 78.3 | 67.5 | 68.5 | 75.1 |
| CSC-R (Du et al., 2025) | MLCIL | | 79.3 | 67.5 | 68.5 | 73.9 | 78.5 | 67.8 | 69.7 | 76.0 |
| PODNet (Douillard et al., 2020) | CIL | | 43.7 | 7.2 | 14.1 | 25.6 | 44.3 | 6.8 | 13.9 | 24.7 |
| oEWC (Schwarz et al., 2018) | CIL | | 46.9 | 6.7 | 13.4 | 24.3 | 44.8 | 11.1 | 16.5 | 27.3 |
| LWF (Li & Hoiem, 2017) | CIL | 0 | 47.9 | 9.0 | 15.1 | 28.9 | 48.6 | 9.5 | 15.8 | 29.9 |
| KRT (Dong et al., 2023) | MLCIL | | 74.6 | 55.6 | 56.5 | 65.9 | 77.8 | 64.4 | 63.4 | 74.0 |
| CSC (Du et al., 2025) | MLCIL | | 78.0 | 64.9 | 66.8 | 72.8 | 78.2 | 65.7 | 67.0 | 75.0 |
| L3A | MLCIL | 0 | **81.5** | **69.7** | **72.5** | **77.6** | **79.9** | **69.5** | **72.9** | **78.8** |

class names.

### 4.1.2. EVALUATION METRICS

Following previous MLCIL works (Dong et al., 2023; De Min et al., 2024), we adopt the mean average precision (mAP) as the primary evaluation metric for each MLCIL phase, and report both the average mAP (the average of the mAP across all phases) and the last mAP (the mAP of the last phase). We also report the per-class F1 score (CF1) and overall F1 score (OF1) for performance evaluation.

### 4.1.3. IMPLEMENTATION DETAILS

We adopt ImageNet-21k pre-trained TResNetM (Ridnik et al., 2021) as our backbone (All compared methods adopt pre-trained TResNetM or ViT-B/16 as the backbone). The batch size is set to 64 for MS-COCO and 256 for PASCAL VOC. In all experimental protocols, we set the regulariza-

tion term $\gamma$ in Equation (8) to 1000, and the buffer layer size to 8192 for MS-COCO and PASCAL VOC.

### 4.2. Comparison Methods

We select several advanced methods in single-label class incremental and multi-label class incremental for comprehensive comparison.

**Baseline methods.** Upper-bound represents the ideal performance result, assuming the model has access to the data from all tasks simultaneously. Finetune (FT) is the simplest baseline method, where the model is trained directly on the current task with catastrophic forgetting.

**CIL methods.** We select several classical CIL methods for comparison, including replay-based approaches such as PODNet (Douillard et al., 2020) and regularization-based methods like oEWC (Schwarz et al., 2018), with the over-

*Table 2.* Comparison of the results between L3A and prompt-based CIL methods on MS-COCO dataset. Data **in bold** represents the **best** results and data underline represents the second-best results.

| Method | Backbone | Param. | MS-COCO B0-C10 | | MS-COCO B40-C10 | |
| --- | --- | --- | --- | --- | --- | --- |
| | | | Avg. mAP | Last mAP | Avg. mAP | Last mAP |
| Upper-bound | | | - | 83.2 | - | 83.2 |
| L2P (Wang et al., 2022) | ViT-B/16 | 86.0M | 73.4 | 68.0 | 73.7 | 71.1 |
| CODA-P (Smith et al., 2023) | | | 74.0 | 65.4 | 73.9 | 67.5 |
| MULTI-LANE (De Min et al., 2024) | | | 79.1 | 74.5 | 78.8 | 76.6 |
| L3A | TResNet-M | 29.4M | **81.5** | **77.6** | **79.9** | **78.8** |

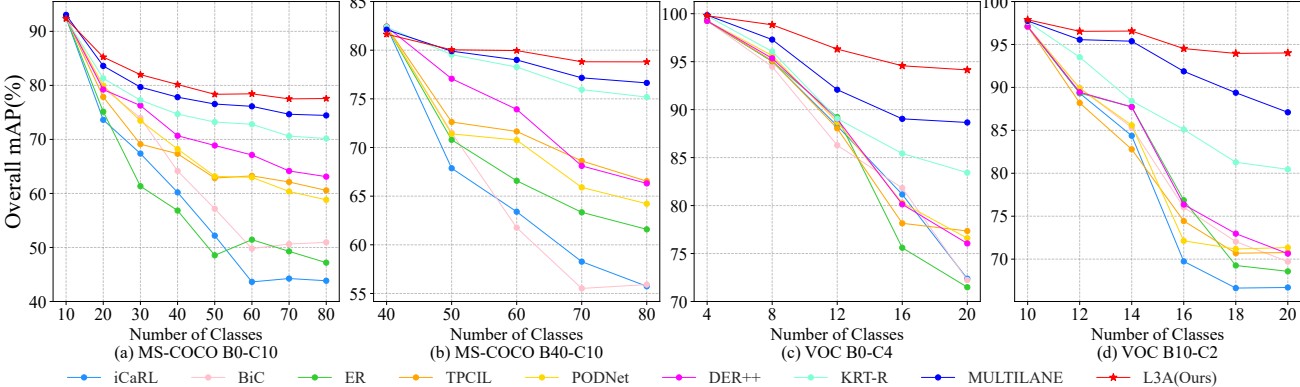

*Figure 3.* Comparison results (mAP%) on MS-COCO and PASCAL VOC datasets under different protocols against competitive methods.

all methods presented in Table 1. Additionally, we include prompt-based methods like L2P (Wang et al., 2022), which leverage the more powerful pre-trained large model ViT-B/16, with their results shown separately in Table 2.

**Multi-label CIL methods.** We select five best-performing MLCIL methods for comparison, including two online replay-based methods PRS (Kim et al., 2020) and OCDM (Liang & Li, 2022), the regularization-based methods KRT and CSC, as well as their replay-based extensions KRT-R (Dong et al., 2023) and CSC-R (Du et al., 2025), and the prompt-based method MULTI-LANE (De Min et al., 2024), which currently represents the SOTA in exemplar-free ML-CIL method.

### 4.3. Comparison Results

#### 4.3.1. RESULTS IN MS-COCO

Table 1 shows the performance comparison on the MS-COCO B0-C10 and B40-C10 benchmarks. L3A demonstrates a significant advantage over other exemplar-free approaches across all metrics, achieving the highest last mAP of 77.6% and 78.8% on the MS-COCO B0-C10 and B40-C10 benchmarks respectively. L3A outperforms the best exemplar-free method CSC in the last mAP by 4.8% and 3.8% on these two benchmarks. Even compared with replay-

based methods, L3A achieves competitive results, outperforming the best replay-based approach CSC-R in the last mAP by 2.8% and 2.1%.

Table 2 compares the performance of L3A with prompt-based methods on the MS-COCO B0-C10 and B40-C10 benchmarks. While prompt-based methods benefit from more powerful backbones with stronger representation capabilities, enabling competitive classification performance without replay, L3A achieves superior results with a less powerful backbone. Specifically, L3A outperforms the best prompt-based MLCIL method MULTI-LANE in the last mAP by 3.1% and 2.2% on the two benchmarks. This demonstrates that our method achieves stronger classification performance with fewer network parameters.

#### 4.3.2. RESULTS IN PASCAL VOC

Table 3 shows the performance comparison on the VOC B0-C4 and B10-C2 benchmarks. L3A achieves results close to the upper-bound, with the last mAP of 94.1% and 94.0% on the VOC B0-C4 and B10-C2 benchmarks respectively, surpassing the current SOTA methods MULTI-LANE by 5.3% and 5.7% on these two benchmarks.

Figure 3 shows the detailed comparison result between L3A and other competitive methods on the four protocols. L3A

achieves more significant performance in every MLCIL phase. Moreover, these performance improvements become increasingly pronounced as the learning phase increases.

The comparison experiments on both MS-COCO and PASCAL VOC datasets demonstrate the remarkable performance of L3A. As an exemplar-free method, L3A achieves competitive advantages over replay-based and prompt-based methods, highlighting its effectiveness in addressing the challenges of MLCIL.

*Table 3.* Comparison of the results on PASCAL VOC dataset. Memory represents the number of replay samples, where 0 indicates that the method is examplar-free. Data **in bold** represents the **best** results and data underline represents the second-best results.

| Method | Memory | VOC B0-C4 | | VOC B10-C2 | |
|---|---|---|---|---|---|
| | | Avg. mAP | Last mAP | Avg. mAP | Last mAP |
| Upper-bound | 0 | - | 94.7 | - | 94.7 |
| FT | | 82.1 | 62.9 | 70.1 | 43.0 |
| iCaRL | | 87.2 | 72.4 | 79.0 | 66.7 |
| BiC | | 86.8 | 72.2 | 81.7 | 69.7 |
| ER | | 86.1 | 71.5 | 81.5 | 68.6 |
| TPCIL | 2/class | 87.6 | 77.3 | 80.7 | 70.8 |
| PODNet | | 88.1 | 76.6 | 81.2 | 71.4 |
| DER++ | | 87.9 | 76.1 | 82.3 | 70.6 |
| KRT-R | | 90.7 | 83.4 | 87.7 | 80.5 |
| CSC-R | | 92.4 | 87.9 | 91.6 | 87.8 |
| CSC | | 90.4 | 85.1 | 89.0 | 83.8 |
| CODA-P | 0 | 90.6 | 84.5 | 90.2 | 85.0 |
| MULTI-LANE | | 93.5 | 88.8 | 93.1 | 88.3 |
| L3A | 0 | **96.7** | **94.1** | **95.6** | **94.0** |

### 4.4. Pseudo-Label Addresses the Label Absence

The PL module addresses the label absence problem by estimating missing labels from past phases. Specifically, it constructs an augmented label set $\hat{Y}_{1:t}$, which combines the true labels from the current phase with pseudo-labels for previous phases. This augmentation enables the analytic learning method in Equation (8) to update $\bar{W}_t$ more accurately by leveraging a complete representation.

Without the label augmentation, the true label set $Y_t$ only contains current phase labels, model progressively loses label information learned in earlier phases. This limitation undermines the precision of the analytic solution for $\bar{W}_t$.

### 4.5. Weighted Analytic Classifier Addresses the Class Imbalance

The WAC addresses the class imbalance problem through a weight fusion strategy. The sample-specific weight $\omega_{t,i}$

for each sample $i$ at phase $t$ is the average of its constituent class-specific weights $v_t^{(k)} = 1/\sqrt{f^{(k)}}$, where $f^{(k)}$ is the frequency of class $k$. This adaptive design of $1/\sqrt{f^{(k)}}$ ensures that rare labels (with small $f^{(k)}$) contribute dominantly to $\omega_{t,i}$, providing smoother scaling and preventing excessive weighting.

Theoretically, this weighting mechanism acts as a soft constraint on the closed-form solution Equation (10), redistributing the residual errors in the least squares objective to penalize misclassifications of rare labels more severely. Crucially, the fusion of class-specific weights ensures a balanced representation of both rare and frequent labels, mitigating the risk of overfitting to dominant classes while maintaining sensitivity to underrepresented ones.

### 4.6. Ablation Study

The ablation study in Table 4 compares the performance of two key modules in L3A. We use the method of freezing the backbone and fine-tuning the classifier as the baseline, the update of the classifier is based on back-propagation. We introduce two additional variants of L3A. (1) L3A w/o WAC: This variant optimizes using only the PL module. (2) L3A w/o PL: This variant optimizes using only the WAC.

Removing the WAC (L3A w/o WAC) results in a significant performance drop, it indicates the model without WAC fails to keep the knowledge of past knowledge. The removal of the PL module (L3A w/o PL) also causes performance degradation. By integrating both two modules, the L3A can effectively acquire new knowledge while retaining previously learned information, with benefits in addressing class imbalance. These results demonstrate both the strengths and effectiveness of the PL module and WAC within the proposed L3A framework.

*Table 4.* Performance comparison with pseudo-label (PL) module and weighted analytic classifier (WAC).

| Model | PL | WAC | COCO B0-C10 | | COCO B40-C10 | |
|---|---|---|---|---|---|---|
| | | | Avg. mAP | Last mAP | Avg. mAP | Last mAP |
| Baseline | ✗ | ✗ | 48.5 | 25.6 | 45.9 | 26.3 |
| (1) w/o WAC | ✓ | ✗ | 70.9 | 56.7 | 77.7 | 72.1 |
| (2) w/o PL | ✗ | ✓ | 81.4 | 77.3 | 79.6 | 78.4 |
| L3A | ✓ | ✓ | **81.5** | **77.6** | **79.9** | **78.8** |

**Regularization term.** As shown in Table 5, we observe that the performance of L3A in a wide range of $\gamma$ value. Setting $\gamma = 1000$ provides good results across MS-COCO benchmarks.

**Buffer layer size.** Table 6 shows that the accuracy of L3A improves with an increase in buffer layer size, but the im-

*Table 5.* The ablation study on regularization term ($\gamma$).

| $\gamma$ | COCO B0-C10 | | COCO B40-C10 | |
|---|---|---|---|---|
| | Avg. mAP | Last mAP | Avg. mAP | Last mAP |
| 0.1 | 80.41 | 76.83 | 79.36 | 78.22 |
| 1 | 80.39 | 76.90 | 79.37 | 78.27 |
| 10 | 80.49 | 76.90 | 79.38 | 78.28 |
| 100 | 80.86 | 77.09 | 79.51 | 78.29 |
| 1000 | **81.38** | 77.56 | **79.89** | **78.75** |
| 10000 | 81.22 | **77.61** | 79.67 | 78.69 |

provement becomes negligible once the size reaches 8196. The size of 8196 is sufficient and a larger size entails more computation on matrix inversion.

*Table 6.* The ablation study on buffer layer size.

| Buffer layer size | COCO B0-C10 | | COCO B40-C10 | |
|---|---|---|---|---|
| | Avg. mAP | Last mAP | Avg. mAP | Last mAP |
| 512 | 79.33 | 75.24 | 77.78 | 76.43 |
| 1024 | 80.47 | 76.52 | 78.89 | 77.67 |
| 2048 | 81.08 | 77.20 | 79.58 | 78.44 |
| 4096 | 81.38 | 77.55 | 79.91 | 78.77 |
| 6144 | **81.43** | **77.60** | 79.94 | 78.75 |
| 8192 | **81.43** | 77.56 | **79.96** | 78.79 |
| 12288 | 81.40 | 77.56 | 79.93 | **78.82** |

**Confidence threshold.** We analyze the effect of the confidence threshold $\eta$ in the PL module. As shown in Table 7, the best performance is consistently observed when $\eta$ is set around 0.7. Motivated by (Dong et al., 2023), we further explore a dynamic thresholding strategy (denoted as *Dynamic* in Table 7) that adaptively adjusts $\eta$ based on the number of pseudo-labels generated in each continual learning phase. However, this method introduces additional hyperparameters and offers no notable performance gain, suggesting that a fixed threshold of 0.7 remains a more effective and practical choice.

*Table 7.* The ablation study on confidence threshold ($\eta$).

| $\eta$ | COCO B0-C10 | | COCO B40-C10 | |
|---|---|---|---|---|
| | Avg. mAP | Last mAP | Avg. mAP | Last mAP |
| 0.55 | 71.25 | 67.21 | 78.30 | 76.12 |
| 0.6 | 78.63 | 74.21 | 79.52 | 78.09 |
| 0.65 | 80.97 | 77.08 | 79.85 | 78.68 |
| 0.7 | **81.45** | 77.57 | **79.91** | **78.77** |
| 0.75 | 81.36 | **77.58** | 79.80 | 78.62 |
| 0.8 | 81.34 | 77.57 | 79.69 | 78.47 |
| *Dynamic* | 81.12 | 76.56 | 79.86 | 78.45 |

**Weighting mechanism.** We evaluate different weighting strategies for the WAC module and find that the proposed class-specific weights $v_t^{(k)} = 1/\sqrt{f^{(k)}}$ formulation achieves the best performance in Table 8. Its smoothing effect effectively alleviates the impact of extreme class imbalance.

*Table 8.* Different weighting mechanism in the WAC module.

| Weighting mechanism | COCO B0-C10 | | COCO B40-C10 | |
|---|---|---|---|---|
| | Avg. mAP | Last mAP | Avg. mAP | Last mAP |
| $1/\sqrt{f^{(k)}}$ | **81.45** | **77.57** | **79.91** | **78.77** |
| $1/f^{(k)}$ | 81.13 | 77.30 | 79.60 | 78.27 |
| $1/(\log(f^{(k)}) + 1)$ | 81.36 | 77.41 | 79.87 | 78.27 |

# 5. Conclusions

In this paper, we focus on the multi-label class-incremental learning (MLCIL). We propose L3A to address label absence, class imbalance, and privacy protection. L3A is an exemplar-free approach comprising two key modules: the pseudo-label (PL) module, which solves label absence by generating pseudo-labels for previously learned classes in current phase samples, and the weighted analytic classifier (WAC) adaptively balances class distributions by introducing sample-specific weights while leveraging a closed-form analytical solution. Experimental results demonstrate that L3A achieves SOTA performance on MS-COCO and PASCAL VOC datasets.

# Acknowledgements

This research was supported by the National Natural Science Foundation of China (62306117, 62406114, 62472181), the Guangzhou Basic and Applied Basic Research Foundation (2024A04J3681), GJYC program of Guangzhou (2024D03J0005), National Key R & D Project from Minister of Science and Technology (2024YFA1211500), the Fundamental Research Funds for the Central Universities (2024ZYGXZR074), Guangdong Basic and Applied Basic Research Foundation (2024A1515010220, 2025A1515011413), and South China University of Technology-TCL Technology Innovation Fund.

# Impact Statement

This paper presents work whose goal is to advance the field of Machine Learning. There are many potential societal consequences of our work, none which we feel must be specifically highlighted here.

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

## A. Proof of Theorem 3.1

*Proof.* The representation of $\bar{W}_{t-1}$ can be written as

$$\bar{W}_{t-1} = (\sum_{i=1}^{t-1} X_i^\top \Omega_i X_i + \gamma I)^{-1} \left[ X_1^\top \Omega_1 \hat{Y}_1 \ \cdots \ X_{t-1}^\top \Omega_{t-1} \hat{Y}_{t-1} \right], \tag{14}$$

and $\bar{W}_t$ can be written as

$$\bar{W}_t = (\sum_{i=1}^{t-1} X_i^\top \Omega_i X_i + \gamma I + X_t^\top \Omega_t X_t)^{-1} \left[ X_1^\top \Omega_1 \hat{Y}_1 \ \cdots \ X_{t-1}^\top \Omega_{t-1} \hat{Y}_{t-1} \ X_t^\top \Omega_t \hat{Y}_t \right]. \tag{15}$$

The autocorrelation matrix in phase $t-1$ can be defined as

$$R_{t-1} = (\sum_{i=1}^{t-1} X_i^\top \Omega_i X_i + \gamma I)^{-1}. \tag{16}$$

We can define $R_t$ from $R_{t-1}$ through Equation (16)

$$R_t = (\sum_{i=1}^{t-1} X_i^\top \Omega_i X_i + \gamma I + X_t^\top \Omega_t X_t)^{-1}$$
$$= (R_{t-1}^{-1} + X_t^\top \Omega_t X_t)^{-1}. \tag{17}$$

According to the Woodbury matrix identity, we have

$$(A + UCV)^{-1} = A^{-1} - A^{-1} U (C^{-1} + V A^{-1} U)^{-1} V A^{-1}.$$

Let $A = R_{t-1}^{-1}, U = X_t^\top, C = \Omega_t$, and $V = X_t$ in Equation (17) , we have

$$R_t = R_{t-1} - R_{t-1} X_t^\top \left( \Omega_t^{-1} + X_t R_{t-1} X_t^\top \right)^{-1} X_t R_{t-1}. \tag{18}$$

Now we can obtain $R_t$ from last phase $R_{t-1}$ and other current phase data (e.g., $X_t$) recursively. The update of the autocorrelation matrix is proved.

Substituting Equation (17) into Equation (15), we have

$$\bar{W}_t = R_t \left[ X_1^\top \Omega_1 \hat{Y}_1 \ \cdots \ X_{t-1}^\top \Omega_{t-1} \hat{Y}_{t-1} \ X_t^\top \Omega_t \hat{Y}_t \right]$$
$$= R_t \left[ R_{t-1}^{-1} \bar{W}_{t-1} \ X_t^\top \Omega_t \hat{Y}_t \right]$$
$$= R_t \left[ (R_t^{-1} - X_t^\top \Omega_t X_t) \bar{W}_{t-1} \ X_t^\top \Omega_t \hat{Y}_t \right]$$
$$= \left[ \bar{W}_{t-1} - R_t X_t^\top \Omega_t X_t \bar{W}_{t-1} \ R_t X_t^\top \Omega_t \hat{Y}_t \right], \tag{19}$$

which completes the proof. $\qquad \square$

