# OpenReview forum: "L3A: Label-Augmented Analytic Adaptation for Multi-Label Class Incremental Learning"
_ICML.cc/2025/Conference — ICML 2025 poster_

### Official Review · Reviewer_jGY3 · 2025-03-11

**Overall Recommendation:** 2

**Summary:**

This work studies the task of Multi-Label Class Incremental Learning (MLCIL), where a model is tasked to incrementally learn to assign multiple labels for each image during each incremental session. The goal is to not forget previously leanred classes while learning novel classes. This task is an extension of class incremental learning.

To tackle MLCIL, the authors proposed Label-Augmented Analytic Adaptation (L3A), an exemplar-free approach without storing past sample. Exemplar-free based approach has advantage of protecting privacy, since no past data will be stored.

L3A has two modules: i)  the Pseudo-Label (PL) module, which generates pseudo-labels for previously learned classes to address label absence, and the ii) Weighted Analytic Classifier (WAC), which uses a closed-form solution with sample-specific weights to mitigate class imbalance.

Experiments on MS-COCO and PASCAL VOC (multi-label, or say tags, benchmarks) shows that L3A achieved SOTA performance.

The main contribution of this paper is an exemplar-free MLCIL solution L3A that avoids storing past samples, making it suitable for privacy-sensitive applications, and provides a theoretical framework for recursive updates of the classifier using an autocorrelation matrix.

**Claims And Evidence:**

Yes.

**Essential References Not Discussed:**

To the best of my knowledge, this paper sufficiently discussed related literature.

**Experimental Designs Or Analyses:**

Yes, the experimental design is fair and well-established.

**Methods And Evaluation Criteria:**

Yes.

**Other Comments Or Suggestions:**

See below.

**Other Strengths And Weaknesses:**

Strengths:
1) The proposed appraoch is, as a whole, novel to a certain degree, with a solid theoretical framework.
2)  The Pseudo-Label (PL) module generates pseudo-labels for previously learned classes, effectively addressing the challenge of label absence in multi-label incremental learning.  However, this has been widely used in examplar-free pipelines.
3) The paper is well-written
4) The performance of the proposed method L3A looks good, consistenly outperforming the compared methods across two benchmarks.

**Questions For Authors:**

Q1: How does the Pseudo-Label (PL) module handle cases where the confidence scores for pseudo-labels are ambiguous (e.g., close to the threshold)? Does the method incorporate any mechanisms to refine or correct pseudo-labels over time?

Q2: The WAC uses a weighting mechanism based on class frequency ((v^{(k)} = 1/\sqrt{f^{(k)}})). Why was this specific weighting function chosen, and how does it compare to other possible weighting schemes (e.g., inverse frequency or log-based weighting)?

Q3 (main):  Do the authors think the two studied benchmarks are sufficient? MS-COCO and PSACAL VOC indeed have multiple labels. However, the labels are mainly object tags, since these images have multiple objects in every single image. In reality, "Multiple-label" should come from different pespectives about the visual content, such as: time of the day, mood, action, color scheme, even noises type (ImageNet-C). Will the proposed method also work in the settings where the multiple-lables from different aspects? In addition, when the labels are from different granularity, such as: {person, dog, cat} is a common granularity, however, the label can also be: {green T-shirt with a dog,  black dots, yellow ear} where the labels are from more fine-grained visual details. How can we prove the proposed method also effective in such scenarios?

Q4: It would be interesting to compare the proposed method with open-vocabulary detectors and taggers.

My main concern is about the insufficient evaluation of the propsoed method.

**Relation To Broader Scientific Literature:**

The main contribution of this paper is an exemplar-free MLCIL solution L3A that avoids storing past samples, making it suitable for privacy-sensitive applications, and provides a theoretical framework for recursive updates of the classifier using an autocorrelation matrix.

**Theoretical Claims:**

Yes.

---

> ### Author Rebuttal · Authors · 2025-04-01
>
> Thank you for your insightful feedback and constructive suggestions. Below, we provide a point-by-point response to your questions and concerns.
>
> ---
> **Q1**: Ambiguous confidence scores and refinement mechanisms in the PL module.
>
> ---
> 1. **Ambiguous confidence scores**, To clarify this issue, we validate our L3A with different thresholds. The results are shown below and we find that the change of results from 0.65 to 0.80 is not significant. This demonstrates that PL could be robust to ambiguous confidence scores since varying thresholds do not have large effect.
>
> **Table1**: The threshold studies on confidence threshold ($\eta$) in COCO dataset.
> | $\eta$ | COCO B0-C10 Avg. mAP | COCO B0-C10 Last mAP | COCO B40-C10 Avg. mAP | COCO B40-C10 Last mAP |
> |-|-|-|-|-|
> |0.55|71.25|67.21|78.30|76.12|
> |0.60|78.63|74.21|79.52|78.09|
> |0.65|80.97|77.08|79.85|78.68|
> |0.70|**81.45**|77.57|**79.91**|**78.77**|
> |0.75|81.36|**77.58**|79.80|78.62|
> |0.80|81.34|77.57|79.69|78.47|
>
> 2. **Self-Correction**, During the learning, pseudo-labels are not treated as static annotations. Instead, they are continuously re-evaluated as new data is introduced. This iterative correction mechanism helps mitigate the impact of early-stage pseudo-labeling errors.
>
> ---
> **Q2**: The choice of different weighting mechanism in WAC module.
>
> ---
> In this paper, we use the 1/sqrt(f^(k)) to compute the weight. To validate its superiority, we compare different weighting mechanisms and weighting strategy using 1/sqrt(f^(k)) achieved the best performance. Due to smoothing effect of 1/sqrt(f^(k)), WAC can maintain a good balance against extreme class imbalances. We will include this study in the manuscript.
>
> **Table2**: Different weighting mechanism in WAC module.
> |Weighting mechanism|COCO B0-C10 Avg. mAP|COCO B0-C10 Last mAP|COCO B40-C10 Avg. mAP|COCO B40-C10 Last mAP|
> |-|-|-|-|-|
> |1/sqrt(f^(k))|**81.45**|**77.57**|**79.91**|**78.77**|
> |1/f^(k)|81.13|77.30|79.60|78.27|
> |1/(log(f^(k))+1)|81.36|77.41|79.87|78.27|
>
> ---
> **Q3**: Analyze the generalization of the L3A method in more diverse multi-label settings.
>
> ---
> Regarding dataset selection, we followed previous multi-label class incremental learning studies [1,2] by using COCO and VOC datasets as benchmarks. The data for multi-label classification needs to contain multi-classes within a single sample, and "multiple objects in every single image" is easy to meet this requirements. We acknowledge that real-world multi-label data can extend beyond object tags to include attributes like time, mood, action, or fine-grained details, but we do not find a dataset having such features while meeting the requirement of multi-classes in a sample (ImageNet-C performs different noises on an image to generate several samples).
>
> For the scenario that multi-labels from different aspects, our L3A can also work in theory. When denoting the fine-grained tags as distinct labels, there is less difference between the labels coming from different objects in the training of WAC. Moreover, the buffer layer could enhance separability of features [3] then help distinguish originally similar features during fine-grained classification.
>
> We are willing to include the validation of L3A on such a dataset, but unfortunately, we did not find a suitable dataset which has different aspects and meets the requirement of multi-classes in a single sample with limited time. Since we may overlook any relevant datasets, could you please designate a suitable dataset? We are open to include the validation accordingly.
>
> ---
> **Q4**: Comparison with open-vocabulary detectors and taggers methods.
>
> ---
> In this paper, we focus on the **class-incremental continual learning**, where the goal is for the model to learn new classes while retaining previously learned knowledge. We need to achieve good performances on both old and new datasets. In contrast, open-vocabulary detectors and taggers typically evaluate on data outside their training distribution (unseen data) to test generalization ability. The motivation is quite different from the continual learning and it may be not appropriate to directly transfer L3A to this task.
>
> [1] S. Dong, et al, "Knowledge restore and transfer for multi-label class-incremental learning," in CVPR 2023.
>
> [2] K. Du, et al, "Confidence self-calibration for multi-label class-incremental learning," in ECCV 2024.
>
> [3] K.-A. Toh, et al, "Between classification-error approximation and weighted least-squares learning," TPAMI, 2008.

---

### Official Review · Reviewer_ybc7 · 2025-03-11

**Overall Recommendation:** 3

**Summary:**

This paper focuses on the multi-label class-incremental continual learning (ML-CICL) task. To address the challenges of missing historical labels and class imbalance in this task, the authors propose an exemplar-free L3A method. Specifically, L3A utilizes a pretrained language model (PLM) to supplement historical labels for samples, mitigating the issue of missing classes. Additionally, it employs weight adaptive calibration (WAC) to obtain sample-adaptive weights, thereby alleviating the class imbalance problem. Experimental results validate the effectiveness of the proposed model.

### update after rebuttal
Based on the author's response, I don't have any other questions, and I will keep my rating.

**Claims And Evidence:**

The paper presents a approach to addressing the challenges in multi-label class-incremental continual learning. The proposed L3A method is supported by logical reasoning, proof reasoning, and the experimental results are provided to validate its effectiveness.

**Essential References Not Discussed:**

No.

**Experimental Designs Or Analyses:**

The experimental design is sound, as the proposed method is evaluated on two public benchmark datasets, COCO and PASCAL VOC, making the results reliable. However, the paper includes limited ablation studies, and a more detailed analysis of different backbone setting would further strengthen the evaluation.

**Methods And Evaluation Criteria:**

The evaluation criteria, including benchmark datasets and performance metrics, appear appropriate for assessing the effectiveness of the method. The experimental setup provides comparisons, supporting the validity of the proposed approach.

**Other Comments Or Suggestions:**

The motivation behind each module is not sufficiently explained. It would be helpful to provide more detailed justifications.

The paper lacks experiments with different backbones. It would be beneficial to include such experiments.

**Other Strengths And Weaknesses:**

WAC is a dynamic weighting-based method. Can it be compared with learnable parameter-based methods? What are the key differences between WAC and such methods?

In other tasks, such as long-tailed classification, there are existing methods to address class imbalance. Would it be possible to compare WAC with these approaches?

Regarding pretrained parameters, L3A uses pretrained weights from ImageNet-21k, but the paper does not specify what pretrained parameters are used for the comparison methods. Does this affect fairness?

In Table 4, why is there a difference of more than 50 between the baseline and the last row? Additionally, the last row should correspond to the L3A method, but it is incorrectly labeled as WAC.

**Questions For Authors:**

None

**Relation To Broader Scientific Literature:**

This method contributes to the broader scientific literature by proposing an exemplar-free approach for multi-label class-incremental continual learning, which reduces memory consumption compared to traditional exemplar-based methods.

**Theoretical Claims:**

The paper provides a theoretical proof for WAC, which is intended to mitigate class imbalance by assigning sample-adaptive weights. The provided proof appears to be logically sound and aligns with established principles in adaptive weighting strategies.

---

> ### Author Rebuttal · Authors · 2025-04-01
>
> Thank you for your constructive feedback and insightful suggestions. Below are our point-to-point responses:
>
> ---
> **W1**: Comparison between WAC and learnable parameter-based methods.
>
> ---
> To clarify, the Weighted Analytic Classifier (WAC) is a recursive analytical learning method, where we address class imbalance using a weighting mechanism. Since analytical learning relies on least-squares solutions, it operates **without loss functions or gradient computations**. This differs from many parameter-based weighted methods using gradient-based optimization, for example: **KRT**, which adds an *Asymmetric Loss* [1] in the loss function to handle class imbalance. **CSC**, which incorporates *maximum entropy regularization* [2] in the loss function.
>
> In contrast, L3A directly computes the **closed-form solution** for network parameters (without backpropagation), requiring less computational time and lower memory than most parameter-based methods. L3A outperforms the existing learnable parameter-based methods in terms of performance as shown in the manuscript and we further validate the efficiency here. L3A can also improve over existing methods in terms of efficiency.
>
> **Table1**: Comparison of methods in terms of runtime and memory usage in COCO B40C10 benchmark.
> |Method|Time (s)|GPU Memory (MB)|
> |-|-|-|
> |**L3A**|1108|5750|
> |KRT (BP-based)|16971|8170|
>
> ---
> **W2**: Lack of comparison with class imbalance methods.
>
> ---
> We have selected several class imbalance methods for comparison in the manuscript, including BiC and PRS, which are originally designed for single-label long-tailed continual learning. However, these methods perform poorly in multi-label scenarios. Another compared method KRT is designed for the class imbalance issue in MLCIL which L3A also outperforms. Detailed experimental results can be found in Table 1 of our paper. If you believe we need to include more relevant techniques, could you please provide some methods we need to include? We are open to include the experiments accordingly.
>
> ---
> **W3**: Unclear pretrained parameters for comparison methods.
>
> ---
> In this paper, all the backbones including TResNetM and ViT-B/16 are pretrained on ImageNet-21k. We will clarify this in the manuscript.
>
> ---
> **W4**: Large performance gap between baseline and mislabeling in Table-4.
>
> ---
> 1. **Performance Gap**, the baseline method in Table-4 of paper is a continuous learning method that does not employ any anti-forgetting method. Only the classifier is trained with a frozen backbone in the baseline and the baseline could suffer from catastrophic forgetting, resulting in a very low Last mAP. With the WAC and PL, L3A can achieve high performance with less forgetting.
>
> 2. **Label Correction**, thank you for pointing the mislabeling problem. We will correct "WAC" to "L3A" in the manuscript.
>
> ---
> **S1**: The motivation behind each module.
>
> ---
> Our paper proposes two main modules, and their motivations are mentioned in the Introduction part. We provide additional clarification:
>
> 1. **To address the challenges of class imbalance and privacy protection**, we introduce a replay-free recursive analytic continual learning method. However, since this approach struggles with class imbalance, we further propose the Weighted Analytic Classifier (WAC), which employs a sample-specific weighting mechanism that adaptively adjusts the importance of samples weights based on class frequency. This correction ensures balanced least-squares optimization
>
> 2. **To tackle the label absence problem**, we adopt a pseudo-labeling strategy (PL module) to generate labels for past tasks. This allows WAC to utilize complete label information during training.
>
> ---
> **S2**: Lack of experiments with different backbones.
>
> ---
> We add experiments with L3A on pre-trained ViT-B/16. The performance of L3A on ViT-B/16 is slightly worse. The reason could be that vanilla ViTs achieve the less local attention than CNN networks when dealing with complex multi-label images (e.g., scenes with many small objects) [3].
>
> **Table2**: Performance results on different backbones
> |Method|backbone|COCO B0-C10 Avg. mAP|COCO B0-C10 Last mAP|COCO B40-C10 Avg. mAP|COCO B40-C10 Last mAP|
> |-|-|-|-|-|-|
> |L3A|TResNetM|81.45|77.57|79.91|78.77|
> |L3A|ViT-B/16|78.32|73.89|73.35|71.66|
>
> [1] T. Ridnik, et al, "Asymmetric loss for multi-label classification," in ICCV 2021.
>
> [2] K. Du, et al, "Confidence self-calibration for multi-label class-incremental learning," in ECCV 2024.
>
> [3] Z. Liu, et al, "Swin Transformer: Hierarchical Vision Transformer using Shifted Windows," in ICCV 2021.

---

### Official Review · Reviewer_x334 · 2025-03-12

**Overall Recommendation:** 2

**Summary:**

This paper proposes Label-Augmented Analytic Adaptation (L3A), an exemplar-free approach without storing past samples, for Multi-Label Class Incremental Learning (MLCIL). It integrates two key modules: the pseudo-label (PL) module implements label augmentation by generating pseudo-labels for current phase samples, and the weighted analytic classifier (WAC) derives a closed-form solution for neural networks. It also introduces sample-specific weights to adaptively balance the class contribution and mitigate class imbalance. Experiments demonstrate the effectiveness of the proposed method. The main contributions of this paper are:

- It proposes L3A, an exemplar-free approach that provides a closed-form solution to address catastrophic forgetting in MLCIL.
- It introduces the PL module to implement label-augmented by generating labels for previously learned classes.
- It introduces the WAC that iteratively updates the classifier by analytic learning and adaptively assigns sample-specific weights.

## update after rebuttal
Thank you for your response. However, in my opinion, the motivation is still somehow confusing. Although the authors indicate some of MLCIL's challenges, the approach they take, especially in tackling class imbalance, is not specially designed for MLCIL's unique challenges. Therefore, I keep my rating.

**Claims And Evidence:**

The claims are supported by evidence. The proposed method's superiority is evident in its consistent outperformance of existing methods.

**Essential References Not Discussed:**

There are no essential references that are missing from the paper.

**Experimental Designs Or Analyses:**

The experimental designs and analyses are overall reasonable. However, the parameter sensitivity analyses are missing. It is not clear how the regularization term $\gamma$ and the buffer layer size affect the final performance.

**Methods And Evaluation Criteria:**

The proposed methods and evaluation criteria make sense for the problem of MLCIL. The proposed L3A is designed to handle the specific challenges of MLCIL, and the evaluation metrics are appropriate for assessing the performance of the proposed method.

**Other Comments Or Suggestions:**

The chosen strategy of the threshold values used for generating pseudo-labels needs to be further refined. Moreover, discussing the limitations of the proposed method and potential future research directions can better facilitate the reader's understanding.

**Other Strengths And Weaknesses:**

### Strengths

- The problem studied in this paper is interesting.

- This paper is well written and in good sharp, which is easy to follow.

- The experimental results are somehow promising.



### Weaknesses

- This paper mainly tackles the **label absence**, **class imbalance**, and **privacy protection** challenges in MLCIL. However, the uniqueness of these challenges in the MLCIL problem is not clear. For example, the class imbalance problem, which can be solved using common weighting means, does not seem to be special in the MLCIL problem. The authors should further explain the uniqueness of these challenges in the MLCIL problem to highlight the significance of the proposed approach.
- The parameter sensitivity analyses are missing. It is not clear how the regularization term $\gamma$ and the buffer layer size affect the final performance.
- From Table 4, one can observe that the effectiveness of PL in the proposed method is not obvious. The authors should analyze the reason for this phenomenon.

**Questions For Authors:**

1. What are the uniqueness of the **label absence**, **class imbalance**, and **privacy protection** challenges in the MLCIL problem?
2. How do the regularization term $\gamma$ and the buffer layer size affect the final performance?
3. The effectiveness of PL in the proposed method is not obvious. The authors should analyze the reason for this phenomenon.

**Relation To Broader Scientific Literature:**

The key contributions of the paper are related to the broader scientific literature. The authors discuss the related works in single-label CIL, multi-label CIL, and analytic continual learning (ACL). They clearly position their work within the context of existing research and highlight the novelty of their approach.

**Theoretical Claims:**

The theoretical claims in the paper are correct. However, Theorem 3.1 is not exactly called a "Theorem", but a computational method.

---

> ### Author Rebuttal · Authors · 2025-04-01
>
> Thank you for your insightful feedback and constructive suggestions. Below we provide point-by-point responses to your comments:
>
> ---
> **W1 & Q1**: Uniqueness of challenges in MLCIL problem.
>
> ---
> Following prior research [1,2], we consider label absence as a unique challenge in MLCIL. Unlike single-label class-incremental learning (SLCIL), MLCIL inherently suffers from missing labels of previous and future labels during training. This increases the difficulty of learning new classes while preserving knowledge of previous ones since learning with only current labels could erase the previous knowledge, making MLCIL fundamentally different from SLCIL in handling label distribution shifts.
>
> For class imbalance, while it is a common issue across machine learning tasks, it has a more severe impact in MLCIL due to catastrophic forgetting. In MLCIL, class distributions are imbalanced not only within each learning phase but also across different incremental stages, where multiple labels coexist per sample, making the imbalance problem even more complex. This leads to the severe under-representation of certain classes as training progresses. As a result, simple class weighting methods often fail to mitigate forgetting effectively in MLCIL, necessitating additional analysis of the class distribution shifts across different incremental learning phases.
>
> For privacy protection, it is a challenge across CIL and existing MLCIL approaches rely on data replaying [1,2], which are incompatible with real-world scenarios when data storage is restricted. This makes exploring exemplar-free methods particularly crucial in MLCIL.
>
> We will further clarify these challenges in the manuscript.
>
> ---
> **W2 & Q2**: Parameter sensitivity analyses of $\gamma$ and buffer layer size.
>
> ---
> We include the parameter analysis as below.
> For $\gamma$, as it increases, the performance of L3A first improves and then declines in **Table 1**. A value of $\gamma$ (1000) achieves consistently good results.
>
> **Table1**: The ablation study on regularization term $\gamma$.
> |$\gamma$|COCO B0-C10 Avg. mAP|COCO B0-C10 Last mAP|COCO B40-C10 Avg. mAP|COCO B40-C10 Last mAP|
> |-|-|-|-|-|
> |0.1|80.41|76.83|79.36|78.22|
> |1|80.39|76.90|79.37|78.27|
> |10|80.49|76.90|79.38|78.28|
> |100|80.86|77.09|79.51|78.29|
> |1000|**81.38**|77.56|**79.89**|**78.75**|
> |10000|81.22|**77.61**|79.67|78.69|
>
> For the buffer size, in **Table 2**, the accuracy of L3A improves with an increase in buffer layer size, but the improvement becomes negligible once the size reaches 4096. The size of 4096 is sufficient and a larger size entails more computation on matrix inversion.
>
> **Table2**: The ablation study on buffer layer size.
> |Buffer layer size|COCO B0-C10 Avg. mAP|COCO B0-C10 Last mAP|COCO B40-C10 Avg. mAP|COCO B40-C10 Last mAP|
> |-|-|-|-|-|
> |512|79.33|75.24|77.78|76.43|
> |1024|80.47|76.52|78.89|77.67|
> |2048|81.08|77.20|79.58|78.44|
> |4096|81.38|77.55|79.91|78.77|
> |6144|**81.43**|**77.60**|79.94|78.75|
> |8192|**81.43**|77.56|**79.96**|78.79|
> |12288|81.40|77.56|79.93|**78.82**|
>
> ---
> **W3 & Q3**: PL module effectiveness analysis.
>
> ---
> We include the analysis as follow.
> "The PL module to address the label absence problem. However, notably, we found that the improvement of using PL upon WAC is less significant in our experiments, while directly using PL upon baseline introduces large improvement. The reason could be that, during the training of WAC, it retains information of previous classes via $R_t$. Then adding the PL is less effective. Nonetheless, the use of the PL module could be still beneficial when using upon both baseline and WAC."
>
> ---
> **S1**: Refine the threshold strategy and analyze limitations.
>
> 1. **Threshold strategy**. To refine this threshold strategy, we explore and validate a dynamic threshold approach, where the threshold is adjusted based on the number of pseudo-labels to be generated in each phase. The result of this approach is denoted as "Dynamic" and is not better than current results. It can add additional hyperparameters of target number of pseudo-labels, which may be unavailable in practice. We will further explore to improve the threshold strategy in future.
>
> **Table3**: Studies on $\eta$
> |$\eta$|COCO B0-C10 Avg. mAP|COCO B0-C10 Last mAP|COCO B40-C10 Avg. mAP|COCO B40-C10 Last mAP|
> |-|-|-|-|-|
> |0.55|71.25|67.21|78.30|76.12|
> |0.6|78.63|74.21|79.52|78.09|
> |0.65|80.97|77.08|79.85|78.68|
> |0.7|**81.45**|77.57|**79.91**|**78.77**|
> |0.75|81.36|**77.58**|79.80|78.62|
> |0.8|81.34|77.57|79.69|78.47|
> |*Dynamic*|81.12|76.56|79.86|78.45|
>
> 2. **Limitations analysis**. L3A utilizes a frozen backbone and updates only the WAC. This could limit the improvement of feature extraction. We will explore to strengthen the backbone's representation to further improve L3A.
>
> [1] Dong, S., et al. Knowledge restore and transfer for multi-label classincremental learning. in ICCV, 2023.
> [2] K. Du, et al, "Confidence self-calibration for multi-label class-incremental learning," in ECCV 2024.

---

### Official Review · Reviewer_VTRd · 2025-03-13

**Overall Recommendation:** 4

**Summary:**

This paper addresses the challenges of multi-label class-incremental learning (MLCIL), specifically label absence, class imbalance, and privacy constraints. The proposed method, L3A, introduces two key modules: 1) Pseudo-Label (PL) Module: Generates pseudo-labels for historical classes using the previous classifier, addressing label absence, 2) Weighted Analytic Classifier (WAC): Uses a closed-form solution with sample-specific weights to balance class contributions, mitigating imbalance. L3A achieves state-of-the-art (SOTA) performance on MS-COCO and PASCAL VOC datasets, outperforming existing exemplar-free and replay-based methods.

**Claims And Evidence:**

The claims (addressing label absence and class imbalance) made in the submission are well-supported as follows,

* Section 4.4 supports the label absence.
* Section 4.5 supports the class imbalance.

**Essential References Not Discussed:**

It seems the references cover all the important ones (just a feeling).

**Experimental Designs Or Analyses:**

Yes, I have checked them. The overall design of experiments and analysis are quite good (e.g., sec. 4.4 and 4.5)

**Methods And Evaluation Criteria:**

Yes, the manuscript employs reasonable methods and evaluation criteria, including common datasets scenarios (MS-COCO and PASCAL VOC), well-recognized metrics (Avg. mAP and Last mAP).

**Other Comments Or Suggestions:**

* Test L3A on datasets with extreme class imbalance (e.g., long-tailed distributions) to further validate its robustness.
* Clarify the buffer layer’s role in feature extraction (e.g., why random linear projections are used).

**Other Strengths And Weaknesses:**

**Strengths**
* Exemplar-Free Privacy: L3A’s analytic updates avoid storing historical data, addressing privacy concerns.
* Comprehensive Evaluation: Rigorous experiments across datasets and protocols validate L3A’s robustness.
* Modular Design: The PL and WAC modules are independently justified and validated.

**Weaknesses**
* The recursive updates of the autocorrelation matrix (Equation 13) may be computationally intensive for large-scale datasets?
* The PL module’s confidence threshold (η) is fixed;

**Questions For Authors:**

* How to choose the weighting strategy?
* How does the claim in Theorem 3.1 helps the multi-label incremnetal learning scenario?

**Relation To Broader Scientific Literature:**

The submission provides a very good discussion of literature. The branches of CL are well partitioned, and can provide reasonable guidance to broader group of readers.

**Theoretical Claims:**

Yes, I have reviewed them. The manuscript includes good theoretical claims, such as the process of optimal solution in Theorem 3.1.

---

> ### Author Rebuttal · Authors · 2025-04-01
>
> Thanks for your review and valuable comments. Here we address your concerns individually as follows:
>
> ---
> **W1**: Recursive updates in Eq. 13 may be computationally intensive.
>
> ---
> The computational cost of our method primarily comes from the matrix inversion in Eq. 13 with the complexity of $\mathcal{O}(d^2N_t + N_t^2d + N_td|C^{1:t}|)$ where `N_t` is the number of samples at stage *t* and `|C^{1:t}|` is the total number of classes up to stage *t*. The complexity scales up linearly with the dataset size and L3A provides training with one-epoch **closed-form solution**. This makes L3A more efficient than existing iterative BP optimization. As shown below, L3A is 10x faster than KRT.
>
> **Table1**: Comparison of methods in terms of runtime and memory usage in COCO B40C10 benchmark.
> |Method|Time(s)|GPU Memory(MB)|
> |-|-|-|
> |**L3A**|**1108**|**5750**|
> |KRT(BP-based)|16971|8170|
>
> ---
> **W2**: Fixed confidence threshold in PL.
>
> ---
> To address this weakness, we explore and validate a dynamic threshold approach, where the threshold is adjusted based on the number of pseudo-labels to be generated in each phase. The results of this approach is denoted as "Dynamic" and is not better. It can add additional hyperparameters of target number of pseudo-labels, which may be unavailable in practice. We will further explore to address the fixed threshold in future.
>
> **Table2**: The threshold studies on confidence threshold ($\eta$) in COCO dataset.
> |$\eta$|B0-C10 Avg. mAP|B0-C10 Last mAP|B40-C10 Avg. mAP|B40-C10 Last mAP|
> |-|-|-|-|-|
> |0.55|71.25|67.21|78.30|76.12|
> |0.6|78.63|74.21|79.52|78.09|
> |0.65|80.97|77.08|79.85|78.68|
> |0.7|**81.45**|77.57|**79.91**|**78.77**|
> |0.75|81.36|**77.58**|79.80|78.62|
> |0.8|81.34|77.57|79.69|78.47|
> |*Dynamic*|81.12|76.56|79.86|78.45|
>
> ---
> **S1**: Test on long-tailed datasets.
>
> ---
> We follow [1] to construct a long-tailed dataset (LT-COCO) by applying a power-law decay sampling method on the COCO dataset then validate on it. Although L3A's performance slightly declined under the long-tailed distribution, it still outperformed other methods, demonstrating its robustness in multi-label continual learning scenarios.
>
> **Table 3**: Comparative results on the LT-COCO dataset (classes are randomly shuffled).
> |Method|LT-COCO B0-C10 Avg. mAP|LT-COCO B0-C10 Last mAP|LT-COCO B40-C10 Avg. mAP|LT-COCO B40-C10 Last mAP|
> |-|-|-|-|-|
> |KRT|74.41|66.89|73.77|70.54|
> |L3A|**76.92**|**72.61**|**74.58**|**73.40**|
>
> ---
> **S2**: Clarify the buffer layer.
>
> ---
> The buffer layer is used to project the model's output to a higher-dimensional space. According to Cover's theorem [2], projecting features into a higher-dimensional space via nonlinear projection functions can improve linear separability. Following ACIL [3], we use a randomly initialized linear projection in this paper and it is simple and effective. We will further clarify this in the text and explore other type of buffer in future.
>
> ---
> **Q1**: Choosing weighting strategy.
>
> ---
> In this paper, we use the 1/sqrt(f^(k)) to compute the weight. To validate its superiority, we compare different weighting mechanisms and weighting strategy using 1/sqrt(f^(k)) achieved the best performance. Due to smoothing effct of 1/sqrt(f^(k)), WAC can maintain a good balance against extreme class imbalances. We will include this study in the manuscript.
>
> **Table4**: Different weighting mechanism in WAC module.
> |Weighting mechanism|COCO B0-C10 Avg. mAP|COCO B0-C10 Last mAP|COCO B40-C10 Avg. mAP|COCO B40-C10 Last mAP|
> |-|-|-|-|-|
> |1/sqrt(f^(k))|**81.45**|**77.57**|**79.91**|**78.77**|
> |1/f^(k)|81.13|77.30|79.60|78.27|
> |1/(log(f^(k))+1)|81.36|77.41|79.87|78.27|
>
> ---
> **Q2**: How Theorem 3.1 helps MLCIL.
>
> ---
> Our work focuses on the MLCIL where we cannot directly apply standard ridge regression (Eq. 10) to optimize all data at once. Here, we extend standard ridge regression into the recursive analytical learning method presented in Theorem 3.1. This allows us to update the classifier through analytical solutions at each incremental learning phase, effectively addressing the MLICL problem.
>
> Also, Theorem 3.1 incorporates two key components of our approach to address issues in MLCIL:
> 1. **PL** (Eq. 3) generates pseudo-labels for past phases, solving the label absence issue.
> 2. **WAC** uses weighted iterative least squares to update classifier, addressing both class imbalance and privacy protection concerns. Theorem 3.1 provides theoretical support for L3A by proving optimal closed-form solution sample-specific weighting for MLCIL.
>
> [1] W. Tong, et al, "Distribution-Balanced Loss for Multi-Label Classification in Long-Tailed Datasets," in EECV 2020.
>
> [2] K.-A. Toh, et al, "Between classification-error approximation and weighted least-squares learning," TPAMI, 2008.
>
> [3] H. Zhuang, et al, "ACIL: Analytic class-incremental learning with absolute memorization and privacy protection," in NeurIPS 2022.

---

> > ### Comment · Reviewer_VTRd · 2025-04-03
> >
> > Thank you for the detailed rebuttal. The authors responded to all concerns, as follows, (i) the efficiency of their work is well justified, with its closed-form solution making it significantly faster than BP-based methods. (ii) the validation on long-tailed setting shows the robustness. (iii) The buffer layer clarification strengthens its theoretical grounding. (iv) The weighting strategy is empirically validated, confirming its effectiveness. (v) Theorem 3.1 is well explained. Thus, I am inclined to accept this work and have raised my score.

---

> > > ### Author Response · Authors · 2025-04-03
> > >
> > > Thank you for taking the time to read our response and increasing your score! We are glad to hear that the response addressed your concern.

---

### Decision · Program_Chairs · 2025-05-01

**Decision:**

Accept (poster)

**Comment:**

In this paper, the authors propose a new task, multi-label class-incremental learning, which extends class-incremental learning to a real-world scenario where each sample may belong to multiple classes. They also introduce a new method, label-augmented analytic adaptation, for this new task.

This paper was reviewed by four expert reviewers. After the rebuttal period, it received mixed ratings. Two reviewers supported this paper, while the remaining two reviewers gave negative ratings. The authors provided many additional results during the rebuttal period. I checked the discussion and believe most of the remaining concerns have been addressed during the rebuttal period. Therefore, I tend to accept this paper. However, Reviewer x334's remaining concerns, such as "this approach is not specially designed for multi-label class-incremental learning's unique challenges", should be discussed in the final version.